# Bone Regeneration Improves with Mesenchymal Stem Cell Derived Extracellular Vesicles (EVs) Combined with Scaffolds: A Systematic Review

**DOI:** 10.3390/biology10070579

**Published:** 2021-06-24

**Authors:** Federica Re, Elena Gabusi, Cristina Manferdini, Domenico Russo, Gina Lisignoli

**Affiliations:** 1Department of Clinical and Experimental Sciences, University of Brescia, Bone Marrow Transplant Unit, ASST Spedali Civili, 25123 Brescia, Italy; domenico.russo@unibs.it; 2Centro di Ricerca Emato-Oncologica AIL (CREA), ASST Spedali Civili, 25123 Brescia, Italy; 3Laboratorio di Immunoreumatologia e Rigenerazione Tissutale, IRCCS Istituto Ortopedico Rizzoli, Via di Barbiano 1/10, 40136 Bologna, Italy; elena.gabusi@ior.it (E.G.); cristina.manferdini@ior.it (C.M.); gina.lisignoli@ior.it (G.L.)

**Keywords:** extracellular vesicles, exosome, mesenchymal stem cell, hydrogels, scaffolds, bone regeneration, tissue inflammation, angiogenesis

## Abstract

**Simple Summary:**

Extracellular vesicles (EVs) have been recently considered one of the main characters for liquid biopsy (biomarkers) and therapeutic application. Particularly, the therapeutic application of EVs is involved in the bone regeneration, thanks to the regulation of immune environments, enhancement of angiogenesis, differentiation of osteoblasts and osteoclasts, and promotion of bone mineralization. In the past 15 years, researchers have focused on the application of EVs derived from different types of mesenchymal stem cells (MSCs) in the field of bone regenerative medicine. This systematic review aims to analyze in vitro and in vivo studies that report the effects of EVs combined with scaffolds in bone regeneration. A methodical review of the literature was performed on PubMed and Embase from 2012 to 2020. Sixteen papers were analyzed; of these, one study was in vitro, eleven were in vivo, and four were both in vitro and in vivo studies. This analysis shows a growing interest in this upcoming field, with overall positive results. Promising in vitro results have been discussed in terms of bone regeneration and pro-angiogenetic processes. The positive in vitro findings were confirmed in vivo, with studies showing positive effects on several critical-size defects. However, some aspects remain to be elucidated, like the different effects induced by EVs and secretome, the most suitable cell source, the EV production protocol and concentration, and the clinical use that may benefit from this new biological approach.

**Abstract:**

Scaffolds associated with mesenchymal stem cell (MSC) derivatives, such as extracellular vesicles (EVs), represent interesting carriers for bone regeneration. This systematic review aims to analyze in vitro and in vivo studies that report the effects of EVs combined with scaffolds in bone regeneration. A methodical review of the literature was performed from PubMed and Embase from 2012 to 2020. Sixteen papers were analyzed; of these, one study was in vitro, eleven were in vivo, and four were both in vitro and in vivo studies. This analysis shows a growing interest in this upcoming field, with overall positive results. In vitro results were demonstrated as both an effect on bone mineralization and proangiogenic ability. The interesting in vitro outcomes were confirmed in vivo. Particularly, these studies showed positive effects on bone regeneration and mineralization, activation of the pathway for bone regeneration, induction of vascularization, and modulation of inflammation. However, several aspects remain to be elucidated, such as the concentration of EVs to use in clinic for bone-related applications and the definition of the real advantages.

## 1. Introduction

In recent years, health problems involving the musculoskeletal system, mainly due to osteoporosis, tumors, and fractures [1], have been widely studied. Different therapeutic approaches are used to induce bone regeneration; however, it has been shown that the physiological processes do not occur in some conditions like critical bone defects, which still remain an unmet clinical need [2] both in orthopedic [3] and maxillofacial [4] surgery. The impossibility to regenerate critical-bone defects is mainly due to the low intrinsic ability to regenerate, as the quantity of bone needed to fill the defect is too large or the bone tissue is compromised by particular situations like osteonecrosis, tumors, or congenital abnormalities [5]. Currently, the gold standard method is the reconstruction with re-vascularized, bone-containing free flaps. These microvascular reconstructions provide optimal results thanks to the vitality of the bone tissue employed. On the other hand, they are technically demanding, requiring high expertise, a considerable time for harvesting and a non-negligible dose of handicraft skills that need several years to be developed [6]. Moreover, the donor site morbidity, although potentially minimized in expert hands, can be considered a further unavoidable drawback of this technique [6]. In this condition, an autologous bone grafting or a cells/scaffold construct implantation is necessary to fill or cover a defect depending on the size and quality of the adjacent soft tissues [7,8].

Bone tissue regeneration procedures have been introduced into clinical practice mainly based on the combination of bone tissue engineering materials with mesenchymal stem cells (MSCs) and/or growth factors [8,9].

MSCs represent the most promising cell population for clinical application in bone and joint diseases, and have been well demonstrated in preliminary clinical studies. MSCs can be derived from different sources like bone marrow and adipose tissue, but also from oral tissues (dental pulp, periodontal ligament, and gingival) [10,11,12]. Particularly, MSCs promote tissue repair thanks to their migration ability to reach injured tissues and by exerting immunomodulatory and trophic effects [13,14]. Moreover, MSCs can be expanded in adequate quantities for potential therapeutic applications, and show a self-renewal capacity and ability to differentiate in vitro into osteoblasts, adipocytes, and chondroblasts [15,16]. However, the efficacy of these approaches may be limited by regulatory issues [17,18]. 

In order to overcome these limitations, researchers (in the past 15 years) have focused on the application of extracellular vesicles (EVs) derived from different types of MSCs in the field of bone regenerative medicine [19,20,21].

EVs have been studied comprehensively in disease therapy and tissue regeneration as they are the primary paracrine executors in signaling communication between cells [22].

EVs derived from MSCs have been shown to exert a compatible regeneration potential compared with MSCs. Thus, EVs are even more appealing than stem cell transplantation as a potential alternative for tissue engineering due to several advantages, including good biosafety, stability, and efficient delivery [23]. EVs circulate in the human body and are present in most biological fluids [24,25].

EVs mainly differ in their dimensions/origin and are classified in exosome, microvesicles (MVs), and apoptotic bodies (ABs) (Figure 1a). Exosomes (<150 nm) are developed from the fusion of multivesicular bodies with the cytoplasmic membrane. MVs (<1mm) are formed through budding of the membrane and shuttle local cytosolic biomolecules. Larger ABs (1–5 µm) are released during the cell apoptotic process and contain cell debris, organelles, and nuclear particulates derived from karyorrhexis [26].

EVs have been proven to play a role in cellular signaling as immunomodulatory messengers [27]. EVs also contribute to improving bone regeneration by increasing angiogenesis [28]. Studies have revealed that EVs maintain the balance of bone metabolism by promoting the differentiation of osteoclasts, osteoblasts, and MSCs [29]. Additionally, EVs participate in bone mineralization, an essential process in bone regeneration [30].

EVs contain different types of biomolecules, such as microRNA (miRNA), short interfering RNA (siRNA), long non-coding RNA (lnRNA), proteins, and cytokines [31,32], and to facilitate their delivery were combined with different scaffolds (Figure 1b).

Among the scaffolds used in bone regeneration, hydrogels can be adapted to have properties closer to the natural extracellular matrix (ECM). In particular, Liu et al. [33] found that a biodegradable hydrogel encapsulating small exosomes can still produce the expected therapeutic effects when applying them directly or near to the treated area [33]. Multi responsive self-healing chitosan based-hydrogels have been extensively studied [34,35,36,37,38,39]. 

However, there are other scaffolds that are currently being used for bone regeneration applications, such as natural polymers (fibrin, hyaluronic acid, and collagen) [40] or synthetic polymers (polyanhydride, polypropylene fumarate (PPF), polycaprolactone (PCL), polyphosphazene, polylactide (PLA) [41], polyether ether ketone (PEEK), and polyglycolide (PGA)) [42], as well as bioactive ceramics of a natural or synthetic origin (coralline, hydroxyapatite, tricalciumphosphate, sulphate, bioactive glass, and calcium silicate) [42,43,44,45,46].

This systematic review aimed to understand the potential of EVs combined with scaffolds as potential therapies in bone regenerative medicine, evidencing the advantages and limits in view of future clinical applications.

## 2. Materials and Methods

### 2.1. Data Source

The use of EVs in both in vitro and in vivo studies for bone regeneration has been systematically reviewed. This search was performed on PubMed and Embase databases from 2012 to 2020, using the following search terms: exosom* OR microvesicle* OR vesicle* OR ectosom* OR secretory* OR embedded-vesicles* OR released vesicle*) AND (bone* OR bone tissue*) AND (regeneration* OR tissue regeneration* OR inflammation* OR Tissue inflammation*) AND (hydrogel* OR biomaterial* OR scaffold*).

### 2.2. Study Selection Process

The screening process of the papers was conducted independently by two reviewers (F.R. and G.L.) by following the PRISMA guidelines. First, the resulting records were screened by title and abstract. After that, the selected manuscripts were assessed considering the in vitro and in vivo studies on the use of EVs and exosomes in bone regeneration. Articles written in other languages or not studying the effect of EVs or not exploiting their potential effect in the bone were excluded. The reference lists of the selected papers were also screened by the reviewers. The flowchart reported in Figure 2 illustrates the systematic review process.

### 2.3. Data Extraction and Synthesis

Relevant data from the selected studies were summarized and analyzed according to the aim of the present manuscript.

Particularly, the cell source of EVs, target cell types, production method, and study design were evaluated [47]. For the in vivo studies, the animal model and the method of bone regeneration were also considered together. In particular, in vitro effects were evaluated in terms of their effect on bone mineralization and their proangiogenic ability; the in vivo effects were evaluated in terms of bone regeneration and mineralization, activation of pathway for bone regeneration, induction of vascularization, and modulation of inflammation.

## 3. Results

One hundred and twenty papers from PubMed and sixty-seven from Embase were found according to the systematic strategy. Sixteen papers were analyzed after the removal of duplicates. Between them, four were in vitro and in vivo, and eleven were in vivo studies. The following paragraphs summarize all of these studies.

### 3.1. In Vitro Studies

Among the five in vitro studies (Table 1), two of them used human periodontal-ligament stem cells (hPDLSCs) [48,49], one used human adipose derived-mesenchymal stem cells (hAD-MSC) [50], one used human gingival stem cell (hGMSC) [51], and one used osteogenic induced human dental pulp stem cell (hDPSC) [47]. Furthermore, one study investigated the effect of exosomes [50] and four investigated the effect of EVs [47,48,49,51]. The mainly reported method to isolate EVs was differential centrifugation, followed by precipitation using commercial kits (three) [48,49,51] and differential centrifugation (two) [47,50]. Only two studies indicated the EV concentrations used [47,50]. Three studies used collagen or PLA with polyethylenimine (PEI) [48,49,51], one used PLA [50], and one used poly(lactic-co-glycolic acid) (PLGA) [47].

#### 3.1.1. Effect on Bone Mineralization

(i.)Diomede et al. [48,51] provided evidence that EVs associated with PEI nanoparticles induced calcium deposition after 6 weeks of culture in basal conditions, with an upregulation of the key genes involved in the pathway of bone differentiation, such as tuftelin 1 (TUFT1), tuftelin 11 (TFIP11), bone morphogenetic proteins (BMP2–BMP), and transforming growth factor (TGFβ) in hPDLSCs and hGMSC. So, PEI associated to EVs synergistically demonstrate a positive effect on cell morphology and gene transcription by increasing the ability to differentiate the osteogenic lineage.(ii.)Pizzicanella J et al. [49] investigated the ability of PEI complexed with hPDLSCs to induce the osteogenic differentiation of hPDLSCs grown on a collagen membrane. In fact, they demonstrated that this system based on collagen membrane plus hPDLSCs and PEI may help to induce bone regeneration.(iii.)Gandolfi et al. [50] showed the ability of exosomes secreted by hAD-MSCs combined with PLA-based scaffolds to trigger the osteogenic commitment of hAD-MSCs, improving their osteogenic properties. Particularly, they used two formulations of PLA+calcium silicates (CaSi)+dicalcium phosphate dihydrate (DCPD), namely: PLA-10CaSi-10DCPD and PLA-5CaSi-5DCPD. Exosomes, encapsulated on the surface of the scaffolds, the improve gene expression of major markers of osteogenesis such as collagen type I (COL1), osteopontin (OPN), osteonectin (ON), and osteocalcin (OCN). The experimental scaffolds enriched with exosomes, in particular PLA-10CaSi-10DCPD, increased the differentiation of MSCs from the osteogenic lineage.(iv.)Benton Swanson W et al. [47] provided strong evidence that osteogenic hDPSCs-derived exosomes facilitate pro-mineralization cues to drive local stem/progenitor cells towards osteogenic lineage on PLLA in vitro.

#### 3.1.2. Effect on Proangiogenic Ability

Pizzicanella J et al. [49] also investigated the ability of PEI complexed with hPDLSCs grown on a collagen membrane to induce the vascularization of bone defects, thanks to its capacity to increase the levels of vascular endothelial growth factor (VEGF) and vascular endothelial growth factor receptor 2 (VEGFR2), which that was shown to play an important role in osteogenesis and bone regeneration. Particularly, PEI-EVs, up-regulating the osteogenic genes and increasing the protein levels of BMP2/4, activated an osteogenic response.

### 3.2. In Vivo Studies

Among the fifteen in vivo studies (Table 2), four included both an in vitro investigation and an animal model study. Four studies were performed on mice [52,53,54,55], ten on rats [48,51,53,56,57,58,59,60,61,62], and one [52] used both mice and rat models. Twelve studies [47,48,49,51,52,55,56,57,58,59,61,62] created an osteochondral defect model and four used [47,52,53,54] subcutaneous implantation. Eight studies investigated the effect of exosomes [47,55,56,57,59,60,61,62], one investigated secretoma [52], five investigated EVs [48,49,51,54,58], and one investigated MVs [53]. Regarding the cell source, three used EVs or secretome from human umbelical cord MSCs (hucMSC) [52,59,62], six from oral MSCs [47,48,49,51,55,58], and four from hBMSCs [53,54,57,61]. Two of them used MSCs derived from human induced pluripotent stem cells (hiPS) [56,60]. Twelve studies used human MSCs [47,48,51,52,53,55,56,57,58,59,60,62] and three used animal MSCs [53,54,61]. One study used a model of rat MSC carrying mutant hypoxia inducible factor-1a (HIF-1a) [61] and one used osteogenic-induced human oral MSCs [47]. A pathological model of osteoporotic rats was also used [60].

The most applied method to isolate EVs was differential centrifugation, followed by ultrafiltration (six) [47,53,58,59,61,62], filtration (one) [52], and precipitation-based commercial kits (seven) [48,49,51,55,56,57,60], while one study performed freeze/thaw cycles [54]. All of the studies combined EVs with a scaffold. The main scaffolds used in these studies were hydrogel with chitosan (two) [55,59], collagen or PLA with PEI (three) [48,49,51], PLA scaffold (one) [47], PCL scaffold (one) [53], tricalcium phosphate (TCP) scaffolds (three) [56,60,61], hyaluronan based-hydrogel (two) [52,62], 3D-printed titanium alloy (one) [57], decalcified bone matrix (DBM) (one) [54], and hydrogel pure matrix (one) [58].

The results of in vivo studies have been summarized according to:

#### 3.2.1. Effect on Bone Mineralization

(i.)Diomede et al. [48] suggested that a commercially available collagen membrane enriched with oral derived stem cells and EVs is capable of inducing bone regeneration.(ii.)It has been shown that engineered EVs with an improvement of the adhesion onto a scaffold could be useful to favor the osteogenic differentiation of MSCs. Particularly, Diomede et al. [51] evaluated the regenerative effects of 3D PLA scaffolds enriched with hGMSCs and complexed with engineered EVs, demonstrating their advantageous use both in vivo and in vitro. EVs were engineered by coating EVs with branched PEI.(iii.)Benton Swanson et al. [47] highlighted how the delivery of the exosomes with a scaffold is able to recruit endogenous cells and stimulates the neogenesis of bone tissue in vivo, without transplantation of the stem cells.(iv.)Not only exosomes, but also secretoma, mainly composed of various growth factors, cytokines, and microRNAs, may affect the differentiation abilities of MSCs as an alternative, as demonstrated by Wang et al. [52]. Particularly, they investigated the effects of secretion factors of hucMSCs on the osteogenesis of hBMSCs both in subcutaneous implantation and in critical-size calvarian defects [52], demonstrating enhanced bone repair.

#### 3.2.2. Effect on Activation of Pathway for Bone Regeneration

(i.)Zhang et al. [56] provided evidence that the exosomes secreted by hiPS and scaffold based on tricalcium phosphate can effectively promote bone repair and regeneration in a rat model of calvarial bone defects through the activation of the phosphoinositide 3-kinases/protein kinase B (PI3K/Akt) signaling pathway on BMSCs [56].(ii.)Cell-free bone regeneration was demonstrated by Zhai et al. [57], who revealed that scaffolds without cells can induce bone regeneration as efficiently as the hMSC-seeded exosome-free scaffolds. Particularly, osteogenic exosomes can be identified from pre-differentiated stem cells and thus used to replace stem cells in tissue regeneration. In fact, exosomes contain upregulated osteogenic miRNAs and thus trigger PI3K/Akt and mitogen-activated protein kinase (MAPK) osteogenic differentiation pathways [57].(iii.)The MAPK pathway was shown to also be activated in hADSCs by hDPSC-EVs as a cell-free biomaterial in a model of the mandibular defects in rat [58].

#### 3.2.3. Effect on Both Bone Regeneration and Vascularization

(i.)Wang et al. [59] produced a hydrogel based on hydroxyapatite, silk fibroin, and glycol chitosan (hydroxyapatite (CHA)/silk fibroin (SF)/glycol chitosan (GCS)/difunctionalized polyethylene glycol (DF-PEG) self-healing hydrogel) with desirable structural and physical properties for bone healing and vehicles of exosomes. Particularly, the combination of the exosomes from hucMSCs and CHA/SF/GCS/DF-PEG, hydrogels could effectively promote the bone healing in Sprague-Dawley rats, with induced femoral condyle defects, by promoting the bone morphogenetic protein-2 (BMP2) deposition, bone collagen deposition, and maturation and enhancing angiogenesis. In this way, Wang et al. [59] demonstrated that hydrogel could become a new type of bone graft material as it has the effect of promoting bone repair, which is more significant after the addition of hucMSC-derived exosomes.(ii.)Critical-sized calvarial defects in an ovariectomized rat model of osteoporosis were induced for repair along with the application of exosomes secreted by MSCs derived from hiPS through enhanced angiogenesis and osteogenesis [60].(iii.)Given the important role played by angiogenesis for bone growth and regeneration, Pizzicannella et al. [49] developed a new construct based on collagen membranes enriched with engineered EVs from hPDLSCs able to promote bone regeneration, as well as the expression of pro-angiogenic factors with consequent vascularization both in vitro and in vivo in rats. EVs were engineered by coating EVs with branched PEI.(iv.)The proangiogenic properties of EVs and hydrogels were also demonstrated by Xie et al. [53], who developed a construct based on MSC-derived microvesicles incorporated into alginate-PCL. These constructs led to increases in vessel formation and tissue-engineered bone regeneration in a subcutaneous bone formation model in nude mice.(v.)DBM with MSC-derived EVs have been demonstrated to have a pro-angiogenic potential and pro-bone regeneration activities, enhancing bone regeneration in a subcutaneous bone formation model in nude mice [54].(vi.)Ying et al. [61] evaluated, for the first time, the role of exosomes carrying mutant hypoxia-inducible factor 1α (HIF-1α), which play an important role in promoting osteogenesis and vascular regeneration, for repairing critical-sized bone defects. HIF-1α-mediated promotion of angiogenesis was also evaluated in a rat model of stabilized fractures by Zhang et al., 2019 [62].

#### 3.2.4. Effect on Inflammation and Cytokines

Another interesting work demonstrated that the association of chitosan hydrogel and dental pulp stem cell-derived exosomes can effectively treat periodontitis, accelerating the healing of alveolar bone and the periodontal epithelium in mice [55] and reducing inflammatory cytokines. The suppression of periodontal inflammation is mediated by macrophages converted from a pro-inflammatory phenotype to an anti-inflammatory phenotype in the periodontium of the mice [53].

## 4. Discussion

Direct cell–cell contact or the transfer of secreted molecules allows intercellular communication, which is an essential hallmark of multicellular organisms. In the last two decades, a third mechanism for communication between cells has emerged, and it involves the intercellular transfer of EVs. EVs are considered to be one of the main characters for liquid biopsy (biomarkers) and therapeutic application [38]. Particularly, the therapeutic application of EVs is involved in the bone regeneration, thanks to the regulation of immune environments, enhancement of angiogenesis, differentiation of osteoblasts and osteoclasts, and promotion of bone mineralization [54].

Various miRNAs and proteins are present in EVs derived from osteoblasts, osteoclasts, osteocytes, monocytes, macrophages, and dendritic cells, and are involved in enhancing or inhibiting the osteogenic activity, as reported by a recent review [63]. In particular, EVs play some role in the calcification of cartilage, bone, and dentin [29]. It has been shown that EV mediated-miRNA is a potential target of high mobility group AT-hook 2 (HMGA2) [64], glycogen synthase kinase-3β/β-catenin [65], or Wnt/β-catenin [66], and is well known to play a role in osteogenic differentiation. Moreover, EVs are also enriched in proteins like tissue non-specific alkaline phosphatase (TNAP), nucleotide pyrophosphatase phosphodiesterase (NPP1/PC-1), annexins (ANX; principally annexins II, V and VI) and phosphatidyl serine (PS) relative to the membranes from which they are derived, matrix metalloproteinases (MMPs), proteoglycan link proteins and actin, a variety of integrins, and PHOSPHO-1 [30], which are all involved in bone remodeling.

The main finding of this review is that EVs associated with different scaffold types could efficiently improve bone regeneration by enhancing angiogenesis through the activation of specific pathways.

In particular, the five in vitro studies [47,48,49,50,51] evidenced positive effects on the induction of bone mineralization, associated with an increase in the specific genes of bone differentiation and the induction of the VEGF angiogenic factor. The 15 in vivo studies [47,48,49,51,52,53,54,55,56,57,58,59,60,61,67] confirmed these effects and evidenced that it is possible to induce bone regeneration by promoting new vessel formation. In fact, the new vessels are fundamental, as the endothelial cells provide nutrients and oxygen necessary for osteogenesis [42].

Angiogenesis plays an important role in bone growth and regeneration [68,69,70], and is also promoted by EVs from both human [49,59,60,67] and rat [53,54,61] MSCs associated with different scaffold types. Interestingly, it has been demonstrated that angiogenesis and osteogenesis was also observed in critical-sized bone defect repair in osteoporotic rats using exosomes secreted from iPSC-MSCs [60], suggesting their positive action also when used in pathological conditions. The important role of HIF-1α for promoting osteogenesis and vascular regeneration has been shown [71,72]. Both Ying et al. [61] and Zhang et al. [62] evaluated, for the first time, the role of exosomes to carry mutant HIF-1a in repairing critical-sized calvarial defects, confirming its important role in the promotion of bone regeneration and neovascularization.

Another feature of EVs carried by different scaffold types is the up-regulation of different osteogenic genes, like RUNX2, OCN, OPN, TUFT1, TFIP11, and COL1A1 [49,51,53], as well as the expression of well-known bone inducing factors like BMP2/4 and TGFβ [49,51]. Vascularization, mineralization, and bone regeneration are strictly linked, and all of these steps are necessary for obtaining correct bone regeneration. It is well known that the induction of ECM followed by mineralization is a fundamental step during bone healing, and the establishment of osteocyte concomitant is associated with the peak expression of genes that are typical markers of mature osteoblasts. These include, but are not limited to, RUNX, bone sialoprotein, OCN, OPN, and BMP pathways, but also local growth factors such as bone TGF-β1/2 [69,70].

Some studies [56,57,58] have focused on the activation of the pathways involved in bone regeneration, such as PI3K/AKT and MAPK signaling pathways. These signaling pathways play important roles in the osteogenesis of the hMSCs and could help to confirm the osteogenic ability of the exosomes. The PI3K/Akt pathway may have a role in the exosome induced pro-osteogenic effects on MSCs, as this signaling cascade has been reported to play important roles in osteoblast differentiation and bone growth. Particularly, an interplay between the PI3K/AKT signaling cascade and BMP-2 gene transcription that regulates osteoblastogenesis has been demonstrated [73]. The MAPK signaling is a key player in bone development and skeletal homeostasis, particularly in osteoblast differentiation [74]. EVs have been found to increase the proliferation of osteoblasts and MSCs through the MAPK pathway [75]. To activate the PI3K/Akt and MAPK signaling pathways, osteogenic exosomes induce osteogenic differentiation by using their cargos, including upregulated osteogenic miRNAs (Hsa-miR-146a-5p, Hsa-miR-503-5p, Hsa-miR-483-3p, and Hsa-miR-129-5p) [45].

As demonstrated by Shen et al. [53], EVs reduce inflammatory cytokines. In fact, exosomes have emerged as potent stimulators of immune responses and as potential biomarkers and therapeutic agents for autoimmune disorders, even if their precise functions and potential in autoimmune diseases are not fully understood [27]. An important role in modulating the phenotype of macrophages has been recently reported using cell-derived exosomes containing miRNAs such as miR-223 [76] and miR-182 [77]. In fact, one of the important mechanisms of the immunomodulatory effects of stem cell exosomes is the exosome-mediated transfer of miRNAs [53]. Among miRNA, Shen et al. [53] focalized on miR-1246, an osteogenic miRNA demonstrating that facilitates the conversion of macrophages from a pro-inflammatory to an anti-inflammatory phenotypes [53], and Zhai et al. [45] also confirmed the presence of other osteogenic miRNAs in the exosome cargos.

It has been reported that no significant effects were observed using free exosome treatment, because of its rapid excretion from the site of application [78]. In fact, exosomes diffused out from the defect rapidly [79]; for this reason, a three-dimensional matrix that can support cell infiltration and vascularization is critical to support tissue neogenesis [80]. Scaffolds are 3D porous substrates promoting the cell-biomaterial interactions, permitting transport of gases, nutrients, and factors for cell survival, proliferation, and differentiation [81].

In these studies, the main scaffolds used are hydrogels [43,46,47,52,53], collagen membrane or PLA with PEI [37,38,39], PLA [40,41], PCL [49], TCP scaffolds [44,48,51], titanium alloy [45], and DBM [50].

Particularly, special attention has been given to hydrogels, as they offer the possibility to generate well-defined 3D biofabricated tissue analogs to the natural ECM, and they have been identified to be proangiogenic during tissue healing [82].

Hydrogels have excellent comprehensive properties and are expected to become a new type of bone graft material, as their effects of promoting bone repair are more significant after the addition of MSC-derived exosomes [59]. Moreover, bone regeneration using hydrogels has been detected starting from 7 days after implantation in rats [62], and generally has been evidenced in 6 weeks after implantation [55,62].

Nevertheless, calcium phosphate based bioceramics are the most widely used osteoinductive materials, including TCP [57,83]. In fact, classical porous β-TCP scaffolds have a good bone conduction performance and bone repair mechanism, which has been evidenced 2 months after the implantation in animal models by Whang et al. [52], Zhang et al. [56], and Qi et al. [60], while this was shown for almost 3 months by Ying et al. [61].

Osteoconductive materials, able to support new bone formation on their surfaces, can also be introduced into non-biological materials (e.g., metal, ceramics, and synthetic polymers) by using various strategies such as coating or composite materials [45].

For example, although titanium is generally considered to be not osteoconductive, bone conduction has been obtained by using an appropriate surface treatment of the titania layer [57,84]. In fact, Zhai et al. [57] described the peculiarities of this kind of material, stating that titanium materials are biocompatible and non-toxic after implantation; they own good mechanical strength to support the bone; and their structures have optimal porosity for cell attachment, migration, and proliferation .

Osteoconductivity of synthetic polymers such as PLA and PCL scaffolds has been realized by developing composite materials with a calcium phosphate coating [50,85]. Interesting results have been obtained using PCL after 1 and 2 months in critical size defects of rat models, where an increase of both bone and vessels formation has been achieved [53].

PEI was often used for the engineering of EVs [48,49,51]. PEI, a complexed nucleic acid, is a well-known polymer useful for promoting the endosomal content release without the need for an additional endosomolytic agent [86]. Moreover, PEI has been demonstrated to have a high affinity to PLA scaffolds, activating this material [48,87]. Both Diomede et al. [48,51] and Pizzicanella et al. [49] evidenced an increase in bone content, mineral contents, and BMP, as well as angiogenesis, 6 weeks after the implantation of PLA-based scaffolds combined with EVs. All these data evidence the positive effects of scaffolds combined with EVs in bone regeneration; however, further studies are necessary to dissect if the mechanism of action is mainly dependent by the scaffold or EV types that were combined. Future research in this field is necessary to understand the therapeutic mechanism of EVs with scaffolds, also considering larger animal models (i.e., sheep) that are closer to humans. This systematic review shows that the most used cell sources are currently oral MSCs and bone marrow MSCs, followed by hucMSCs and hiPS. However, there is a lack of information about the differences between the vesicles derived from different stem source, as well as among the EVs types used, their size, and the isolation procedures, which clearly evidence the limitations of these studies. The most investigated EV types (Figure 1a) are exosomes, followed by EVs and secretoma. The studies show that both EVs and exosomes exert similar osteogenic and mineralization effects, leaving the question on the most suitable approach still open. A critical aspect has been represented by the isolation procedures, as there is no standardized method to isolate EVs, and therefore different products could be used that could represent a critical issue for clinical applications.

Finally, the proper dosage of EVs is another key factor to be considered. Generally, 1, 20, and 50 µg/mL are used, but not all works indicated their concentrations. Moreover, the time points evaluated for bone regeneration in rat and mice models was variable, ranging from 1 to 3 months.

Nevertheless, most of the studies reported a dose-dependent tissue repair of the EVs from MSCs, and particularly high concentrations in vitro (10–100 μg/mL) are preferred [88,89].

However, the lack of standardization and thus the presence of heterogeneous products does not allow for identifying the best EVs concentration for an optimal effect in terms of bone regeneration. Further efforts should investigate the protocols to optimize the EV production and concentration for bone-related applications and for the definition of the related advantages.

These promising results support the potential of this new biological approach, opening new future perspectives for stem cell-based therapy. However, it is crucial to have further efforts to standardize the reporting of the methodology. In this light, new research is required for the identification of the proper cell source; the best preparation protocol for EV isolation; and the most suitable scaffolds in bone regeneration, also in larger animal models in order to facilitate clinical translationality.

## 5. Conclusions

Increasing interest towards EVs as a cell free bone regeneration method has been underlined in this systematic review, with overall positive findings. Promising in vitro results have been discussed in terms of bone regeneration and pro-angiogenetic processes, as summarized in the cartoon (Figure 3). The positive in vitro findings were confirmed by in vivo studies, showing positive effects on several critical-size defects. However, some aspects remain to be elucidated, like the different effects induced by EVs and secretome, the most suitable cell source, the EVs production protocol and concentration, and the clinical use that may benefit from this new biological approach.

## Figures and Tables

**Figure 1 biology-10-00579-f001:**
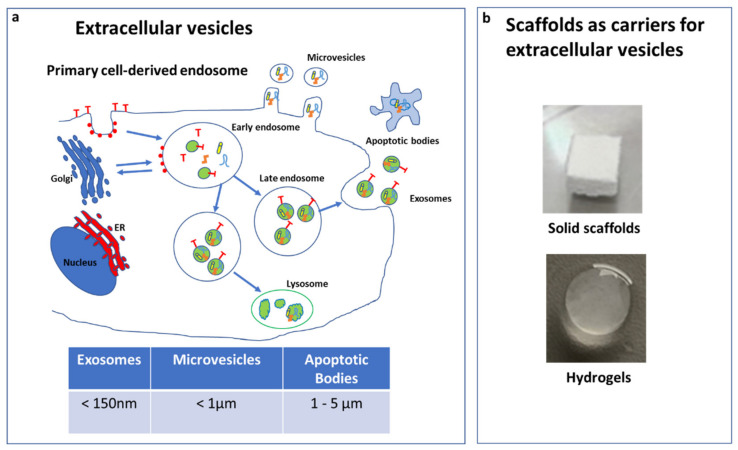
Extracellular vesicle (EV) types and dimension (**a**). Scaffolds used as carriers for EVs (**b**).

**Figure 2 biology-10-00579-f002:**
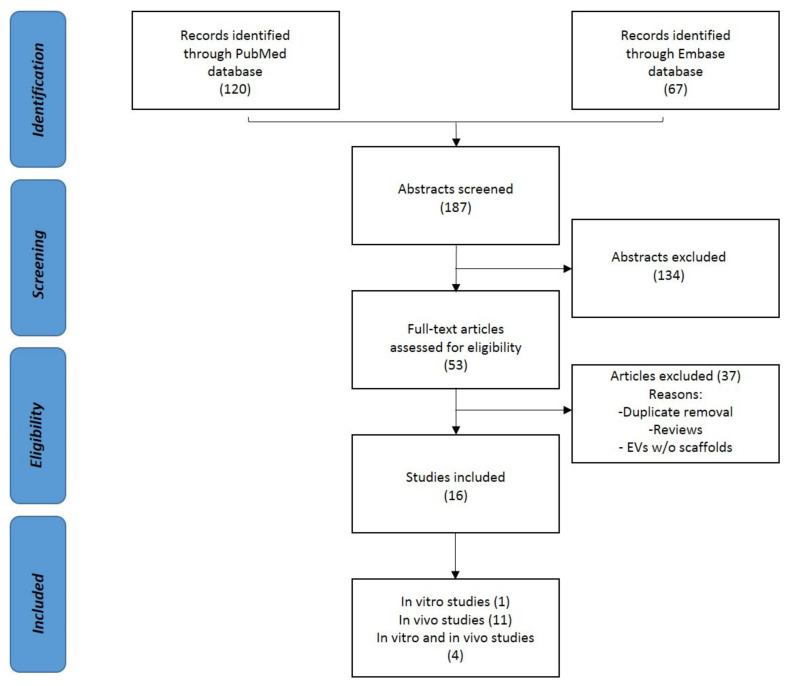
PRISMA (Preferred Reporting Items for Systematic Review and Meta-Analysis) flowchart of the systematic literature review.

**Figure 3 biology-10-00579-f003:**
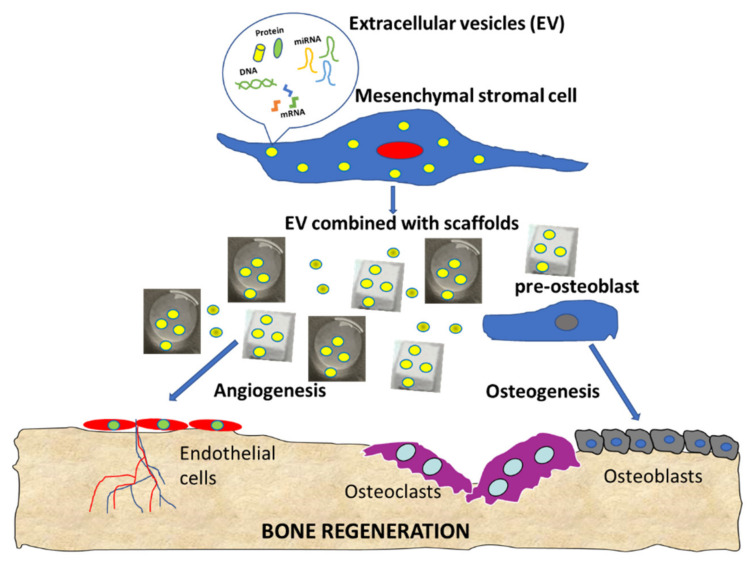
Summary cartoon of EV functions in bone regeneration.

**Table 1 biology-10-00579-t001:** EVs combined with scaffolds used to promote bone regeneration: studies in vitro.

Cells Derived EV	EV Carrier	EV Concentrations	Targets	Main Results	Reference
Human periodontal-ligament stem cells (hPDLSCs)	Collagen/Polyethylenimine (PEI)	Not indicated	Osteogenic differentiation induction of hPDLSCs grown on Collagen membrane	Increase of mineralized matrix and osteogenic genes (TGFB1, MMP8,TUFT1, TFIP11,BMP2, and BMP4)	Diomede F et al., 2018 [48]
Human gingival stem cell (hGMSC)	Polylactide (PLA)/Polyethylenimine (PEI)	Not indicated	Osteogenic differentiation induction of hGMSC grown on Collagen membrane	Increase of mineralized matrix and osteogenic genes (TGFBR1, SMAD1, MAPK1, MAPK14, RUNX2, and BMP2/4)	Diomede F et al., 2018 [51]
Human periodontal-ligament stem cells (hPDLSCs)	Collagen/Polyethylenimine (PEI)	Not indicated	Osteogenic differentiation induction of hPDLSCs grown on Collagen membrane	Increase of osteogenic (RUNX2, COL1A1, and BMP2/4) and angiogenic (VEGF and VEGFR2) genes	Pizzicanella J et al., 2019 [49]
Human adipose derived-mesenchymal stem cells (hAD-MSC)	Two formulations of Polylactic acid(PLA)+calcium silicates (CaSi)+dicalcium phosphate dihydrate (DCPD): PLA-10CaSi-10DCPD and PLA-5CaSi-5DCPD	5 × 10^10^ /cm^2^	Osteogenic differentiation of hAD-MSC	PLA-10CaSi-10DCPD increased Collagen type 1, osteopontin, osteonectin, and osteocalcin runx osteogenic genes	Gandolfi MG et al., 2020 [50]
Osteogenic induced human dental pulp stem cell (hDPSC)	Poly(lactic-co-glycolic acid) (PLGA) and poly(ethylene glycol) (PEG)	2000 µg/ml	Osteogenic differentiation induction of hDPSC grown on Poly (L-lactic-acid) (PLLA)	Increase mineralization	Swanson BW et al., 2020 [47]

Results of in vitro studies were summarized according to:

**Table 2 biology-10-00579-t002:** EVs combined with scaffold used to promote bone regeneration: studies in vivo.

Cells Derived EV	EV Carrier	Species	Target	Time Points	Main Results	Reference
Human periodontal-ligament stem cells (hPDLSCs)	Collagen/Polyethylenimine (PEI)	Male Wistar rat	Healing of frontoparietal region (1 cm) treated with hPDLSCs grown on Collagen membrane+PEI EV	6 weeks	Increase of BMP2 and BMP4	Diomede F et al., 2018 [48]
Human gingival stem cell (hGMSC)	Polylactide (PLA)/ Polyethylenimine (PEI)	Male Wistar rat	Healing of frontoparietal region treated with hGMSC grown on collagen membrane+PEI EV	6 weeks	Increase bone regeneration and angiogenesis	Diomede F et al., 2018 [51]
Osteogenic induced Human dental pulp stem cell (hDPSC)	Poly(lactic-co-glycolic acid) (PLGA) and poly(ethylene glycol) (PEG)	8–10 week old C57BL/6 mice	Subcutaneous implantation -Healing of critical-size calvarial defect	8 weeks	Increase bone formation	Benton Swanson W et al., 2020 [47]
Human umbilical cord mesenchymal stem cells (hucMSC)	Hydroxyapatite/tricalcium phosphate (HA/TCP)	Male nude mice	Subcutaneous implantation	8 weeks	Increase of osteoid matrix and osteocalcin	Wang K-X et al., 2015 [52]
Human umbilical cord mesenchymal stem cells (hucMSC)	2% Hyaluronic acid hydrogel	Male Sprague Dawley rats	Healing of critical-size calvarial defect	8 weeks	Increase bone regeneration	Wang K-X et al., 2015 [52]
Human induced pluripotent stem cells (hiPSCs)	Porous β-Tricalcium phosphate (TCP)	Sprague Dawley rats	Healing of critical-size calvarial defect	8 weeks	EV dose dependent increase in bone formation; area osteocalcin positive	Zhang J et al., 2016 [56]
Osteogenic induced Human (hMSC)	3D-printed titanium alloy	Male 5–6 weeks old Sprague Dawley rats	Healing of radial bone defect	12 weeks	Increase bone regeneration	Zhai M et al., 2020 [57]
Human dental pulp stem cell (hDPSC)	Hydrogel PuraMatrix^®^	Male Wistar rat	Healing of mandibular defect	6 weeks	Increase bone regeneration via MAPK pathway	Jin Q et al., 2020 [58]
Human umbilical cord mesenchymal stem cells (hucMSC)	Coralline hydroxyapatite (CHA)/silk fibroin (SF)/glycol chitosan (GCS)/ difunctionalized polyethylene glycol (DF-PEG)	Sprague-Dawley rat	Healing of femoral condyle defect	30, 60, and 90 days	Increase of bone volume, mineral contents, bone morphogenic protein 2 (BMP2), and angiogenesis	Wang L et al., 2020 [59]
Human induced pluripotent stem cells (hiPSCs)	Porous β-Tricalcium phosphate (TCP)	Mature female Sprague Dawley rats	Healing of critical-size calvarial defect in osteopenic animal model	8 weeks	Increase of osteogenesis and angiogenesis	Qi X et al., 2016 [60]
Human periodontal-ligament stem cells (hPDLSCs)	Collagen/ Polyethylenimine (PEI)	Male Wistar rat	Healing of frontoparietal region (1 cm) treated with hPDLSCs grown on Collagen membrane+PEI EV	6 weeks	High integration and bone regeneration Overexpression of angiogenic genes (VEGFA and VEGFR2)	Pizzicanella J et al., 2019 [49]
Rat mesenchymal stem cells (MSC)	Alginate-polycaprolactone (PCL)	4 week old male nude mice	Subcutaneous implantation	1 and 2 months	Increase of bone formation and enhancement of vessel formation	Xie H et al., 2016 [53]
Osteogenic induced Rat mesenchymal stem cell (MSC)	Decalcified bone matrix	4 week old male nude mice	Subcutaneous implantation	1 and 2 months	Increase bone formation and vascularization	Xie H et al., 2017 [54]
Rat bone marrow mesenchyme stem cells carry mutant HIF-1α (BMSC- HIF-1α)	Porous β-Tricalcium phosphate (TCP)	12 weeks mature Sprague Dawley rats	Healing of critical-size calvarial defect	12 weeks	Increase bone regeneration and neovascularization	Ying C et al., 2020 [61]
Human umbilical cord mesenchymal stem cells (hucMSC)	Hyaluronan based HyStem-HP hydrogel	12 weeks-old male Wistar rat	Healing of fracture femur	7, 14, 21, and 31 days	Increase bone regeneration and angiogenesis	Zhang Y et al., 2019 [62]
Dental pulp stem cells (DPSC)	Chitosan hydrogel	Male C57BL/6J	Healing of alveolar bone	4 weeks	Suppression of periodontal inflammation and modulation of immune response	Shen Z et al., 2020 [55]

## Data Availability

Not applicable.

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
