# Peer review of "Bone Regeneration Improves with Mesenchymal Stem Cell Derived Extracellular Vesicles (EVs) Combined with Scaffolds: A Systematic Review"

_biology, 2021, doi:10.3390/biology10070579_

Round 1

Reviewer 1 Report

This review of the use of extracellular vesicles for bone engineering and regeneration comprises data published on the use of EV in in vitro and in vivo assays. In such sense is similar to other reviews that have been published from 2019 to date (at least 10 reviews have been published on the field since 2019, This review is contrasting with the others because it shows the different effects obtained by the use of EV, on the production of growth factors, cytokines, ECM components and bone proteins. It should be improved by carrying a comparison between the different scaffolds used for the implantation of the EV into the lesional tissue and perhaps whether authors detect the advantage on the use of some of the cited scaffolds, besides the preference detected in the use of such materials. There exists a review  in which a similar PRISMA analysis was made for the clinical use of EV derived from mesenchymal stem cells, but in this case authors approached to clinical results obtained by the use of different material (Tan et al., 2020, Materials Today Bio 7: 100067). Therefore authors should improve the discussion of their review in order to include a contribution on the perspectives in the field and which are the promising materials that could be applied for bone reconstruction/engineering, instead of citing differences in the obtained results.

Author Response

Thanks for the comment and the advice. Discussion has been improved following the suggestions of the Reviewer. Please see page 10-11 lines 364-368, 370-373, 379-382, 384-387 and 392-399. We hope that this satisfies the Reviewer.

Reviewer 2 Report

The article by Re et al "Bone Regeneration improves with Mesenchymal Stem Cell de-2 rived Extracellular Vesicles (EVs) combined with scaffolds: A 3 Systematic Review" is well-written describing the potential role of stem-cells secreted EVs and their therapeutic applications. Here are my few comments:

  1. The authors should discuss the significance of the present work and how it will inspire future research in this field.
  2. I would like to recommend adding some flow chart or schematic illustration explaining in detail molecular mechanism of EVs-related tissue enginnering applications by Systematic database search.

Author Response

1. The authors should discuss the significance of the present work and how it will inspire future research in this field. Prospettive future

We thank you the Reviewer for his comments. A brief statement explaining future perspectives has been added to the discussion stressing what has already been added to the conclusions: “These promising results support the potential of this new biological approach, opening new future perspectives for stem cells-based therapy. However, it would be crucial further efforts to standardize the reporting of the methodology. In this light, new researches are required for the identification of the proper cell source, the best preparation protocol for EVs isolation and the most suitable scaffolds in bone regeneration also in larger animal models in order to facilitate clinical translationality”. Please see page 11 lines 421-426.

2. I would like to recommend adding some flow chart or schematic illustration explaining in detail molecular mechanism of EVs-related tissue enginnering applications by Systematic database search.

We thank the Reviewer for his advice. We have preferred to add in the discussion a statement explaining the molecular mechanism of EVs-related to bone tissue engineering applications: “Various miRNAs and proteins are present in EVs derived from osteoblasts, osteoclasts, osteocytes, monocytes, macrophages and dendritic cells are involved in enhancing or inhibiting osteogenic activity as reported by a recent review [63]. Particularly EVs, play some role in the calcification of cartilage, bone and dentin [29] . It has been shown that EVs mediated-miRNA are potential target of high mobility group AT-hook 2 (HMGA2)[64] or glycogen synthase kinase-3β/β-catenin [65] or Wnt/β-catenin[66] well known to play a role in osteogenic differentiation. Moreover, EVs are also enriched in proteins like tissue non-specific alkaline phosphatase (TNAP), nucleotide pyrophosphatase phosphodiesterase (NPP1/PC-1), annexins (ANX; principally annexins II, V & VI) and phosphatidyl serine (PS) relative to the membranes from which they are derived, matrix metalloproteinases (MMPs), proteoglycan link proteins and actin, a variety of integrins and PHOSPHO-1 [30] all involved in bone remodeling”. Please see page 9 lines 285-296.

Reviewer 3 Report

Excellent review about MSCs and EVs as well as their exosomes.  It is particularly of interest about potential application of EVs and and exosomes in vivo to enhance bone repair and regeneration.  The authors could expand on their review by explaining more about the mechanisms by which EVs and exosomes generated by MSCs  enhance bone formation and repair or regeneration.  The authors have not listed potential contents of EVs, Exosome or secretome of MSCs that contribute to bone regeneration.

Author Response

We thank you the Reviewer for his comments. A statement explaining the contents of EVs that contribute to bone regeneration has been added to the discussion: “Various miRNAs and proteins are present in EVs derived from osteoblasts, osteo-clasts, osteocytes, monocytes, macrophages and dendritic cells are involved in enhancing or inhibiting osteogenic activity as reported by a recent review [63]. Particularly EVs, play some role in the calcification of cartilage, bone and dentin [29]. It has been shown that EVs mediated-miRNA are potential target of high mobility group AT-hook 2 (HMGA2)[64] or glycogen synthase kinase-3β/β-catenin [65] or Wnt/β-catenin[66] well known to play a role in osteogenic differentiation. Moreover, EVs are also enriched in proteins like tissue non-specific alkaline phosphatase (TNAP), nucleotide pyrophosphatase phosphodiesterase (NPP1/PC-1), annexins (ANX; principally annexins II, V & VI) and phosphatidyl serine (PS) relative to the membranes from which they are derived, matrix metalloproteinases (MMPs), proteoglycan link proteins and actin, a variety of integrins and PHOSPHO-1 [30]all involved in bone remodeling”. Please see page 9 lines 285-296.
